# Low PCSK-9 levels Are Associated with Favorable Neurologic Function after Resuscitation from out of Hospital Cardiac Arrest

**DOI:** 10.3390/jcm9082606

**Published:** 2020-08-11

**Authors:** Anne Merrelaar, Nina Buchtele, Christoph Schriefl, Christian Clodi, Michael Poppe, Florian Ettl, Harald Herkner, Bernd Jilma, Michael Schwameis, Christian Schoergenhofer

**Affiliations:** 1Department of Emergency Medicine, Medical University of Vienna and Austria, 1090 Vienna, Austria; anne.merrelaar@meduniwien.ac.at (A.M.); christoph.schriefl@meduniwien.ac.at (C.S.); christian.clodi@meduniwien.ac.at (C.C.); michael.poppe@meduniwien.ac.at (M.P.); florian.ettl@meduniwien.ac.at (F.E.); harald.herkner@meduniwien.ac.at (H.H.); 2Department of Internal Medicine I, Medical University of Vienna and Austria, 1090 Vienna, Austria; nina.buchtele@meduniwien.ac.at; 3Department of Clinical Pharmacology, Medical University of Vienna and Austria, 1090 Vienna, Austria; bernd.jilma@meduniwien.ac.at (B.J.); christian.schoergenhofer@meduniwien.ac.a (C.S.)

**Keywords:** inflammation mediators, cardiac arrest, cardiopulmonary resuscitation, critical care outcomes, emergency medicine, lipid metabolism

## Abstract

Endotoxemia after cardiopulmonary resuscitation (CPR) is associated with unfavorable outcome. Proprotein convertase subtilisin/kexin type-9 (PCSK–9) regulates low-density lipoprotein receptors, which mediate the hepatic uptake of endotoxins. We hypothesized that PCSK–9 concentrations are associated with neurological outcome in patients after CPR. Successfully resuscitated out-of-hospital cardiac arrest patients were included prospectively (*n* = 79). PCSK–9 levels were measured on admission, 12 h and 24 h thereafter, and after rewarming. The primary outcome was favorable neurologic function at day 30, defined by cerebral performance categories (CPC 1–2 = favorable vs. CPC 3–5 = unfavorable). Receiver operating characteristic curve analysis was used to identify the PCSK–9 level cut-off for optimal discrimination between favorable and unfavorable 30-day neurologic function. Logistic regression models were calculated to estimate the effect of PCSK–9 levels on the primary outcome, given as odds ratio (OR) and 95% confidence interval (95%CI). PCSK–9 levels on admission were significantly lower in patients with favorable 30-day neurologic function (median 158 ng/mL, (quartiles: 124–225) vs. 207 ng/mL (174–259); *p* = 0.019). The optimally discriminating PCSK–9 level cut-off was 165 ng/mL. In patients with PCSK–9 levels ≥ 165 ng/mL, the odds of unfavorable neurological outcome were 4.7-fold higher compared to those with PCSK–9 levels < 165 ng/mL. In conclusion, low PCSK–9 levels were associated with favorable neurologic function.

## 1. Introduction

Proprotein convertase subtilisin/kexin type 9 (PCSK–9) regulates the expression of low-density lipoprotein (LDL) receptors (-R) on hepatocytes [1]. PCSK–9 binds to the LDL-R and facilitates the intracellular degradation of the receptor. Thereby, it reduces hepatic LDL-R expression, LDL uptake and consequently increases circulating LDL [2]. The discovery that people with gain-of-function mutations have hypercholesterolemia [3] and those with loss of function mutations have low levels of LDL and a lower risk to develop coronary events in spite of the presence other non-lipid-related risk factors [4] resulted in the development of anti-PCSK–9 antibodies. Until now, two anti-PCSK–9 antibodies have been marketed: in the pivotal trials, alirocumab and evolocumab both demonstrated potent lipid-lowering effects and both reduced the risk to develop cardiovascular events in risk populations [5,6]. Additionally, a recent meta-analysis provides evidence that treatment with one of the two marketed monoclonal PCSK–9 antibodies, alirocumab and evolocumab, significantly improves lipid profiles of patients and reduces the risk of non-fatal major adverse cardiovascular events [7].

During infections bacterial endotoxins are bound to lipoproteins and cleared from circulation by the LDL–R, which was demonstrated for both lipopolysaccharides (LPS) and lipoteichoic acid [8,9]. In this context, a negative effect of high concentrations of PCSK–9 on the clearance of lipopolysaccharide and lipoteichoic acid was demonstrated in vitro [10,11]. Moreover, Levels et al. demonstrated that infusion of 2 ng/kg bodyweight LPS decreased total cholesterol concentrations significantly and within few hours [12]. Pajkrt et al., showed that infusion of recombinant human high-density lipoproteins abolished the activation of coagulation after infusion of LPS, further indicating the connection of lipoproteins and bacterial toxins [13].

During experimental endotoxemia, PCSK–9 knock-out mice produced significantly lower levels of pro-inflammatory cytokines, including interleukin-6 (IL–6) and tumor necrosis factor–α (TNF–α). This effect was also translatable to healthy volunteers: those with PCSK–9 loss-of-function mutations had significantly lower levels of pro-inflammatory cytokines TNF–α and IL-6 during experimental endotoxemia compared to healthy volunteers with PCSK–9 gain-of-function mutations [14]. In septic shock patients with loss-of function mutations, 28–day survival was significantly improved compared to those with gain-of-function mutations or without loss-of-function mutations [14]. In agreement with this observation PCSK–9 levels were associated with organ failure [11]. It is noteworthy that while inflammation itself increases PCSK–9 levels, acute endotoxemia may decreased them, making interpretation of these levels complicated [2,15]. Only recently, some authors reported that for patients suffering from bacteremia [16], from sepsis or from septic shock [17] lower levels of PCSK–9 were associated with mortality. Thus, these results seem to contradict other studies at first. However, findings of a human endotoxemia trial could provide a possible explanation: the infusion of 2 ng/kg LPS in healthy volunteers decreased PCSK–9 levels by approximately 40% before they returned to baseline [15]. Thus, one could conclude that sepsis or septic shock with long-term presence or repetitive bouts of bacterial toxins in circulation, e.g., from a non-resolved site of infection, could therefore result in decreased PCSK–9 levels and are likely to be associated with high mortality. The connection of PCSK–9, lipoprotein levels and infections is intriguing and the first randomized trials that investigate the effects of PCSK–9 antibodies on sepsis are currently being undertaken (NCT03869073).

During cardiopulmonary resuscitation (CPR) intestinal permeability increases enabling bacterial translocation from the gut and causing endotoxemia [18]. These endotoxins contribute to the inflammatory response during post-cardiac arrest syndrome (PCAS), which was associated with unfavorable patient outcome [19]. Moreover, recent experimental data suggest that PCSK–9 could be involved in the release of pro-inflammatory cytokines by macrophages after hypoxia and reperfusion injury. The authors even suggest that PCSK–9 could alleviate the activation of the nuclear factor “kappa-light-chain-enhancer” of activated B-cells (NFKB) pathway after experimental myocardial ischemia [20]. Furthermore, PCSK–9 also interacts with the oxidized–LDL–receptor (LOX–1), which is a central regulator of inflammation, atherogenesis and apoptosis of endothelial cells. PCSK–9 promotes LOX–1 expression and uptake of oxidized LDL and consequently contributes to a pro-inflammatory condition [21]. Thus, inhibition of PCSK–9 may therefore exert anti-inflammatory and endothelium-protective effects [22].

Based on these data, we hypothesized that PCSK–9 levels may impact on PCAS by alleviating the inflammatory response caused by (i) influencing the detoxification of translocated bacterial toxins and (ii) altering the ischemia-reperfusion induced release of pro-inflammatory cytokines of macrophages and thus be associated with patient outcomes after CPR.

## 2. Materials and Methods

The independent review board of the Medical University of Vienna approved the study, which was performed in accordance with the Declaration of Helsinki. A waiver was obtained for informed consent at admission, and patients were informed of their study participation on regaining consciousness.

Adults (>18 years) with out-of-hospital cardiac arrest of presumed cardiac cause who had achieved the return of spontaneous circulation (ROSC) at admission to the intensive care unit (ICU) of the Department of Emergency Medicine at the Medical University of Vienna, were prospectively included in this study. All patients suffered from acute coronary syndrome and underwent percutaneous coronary intervention after admission. No-flow intervals were defined as the time from collapse to start of CPR, low-flow time as the time from CPR start to sustained ROSC and sustained ROSC as recovery of spontaneous circulation for more than 20 min. These intervals were established by immediate, structured interviews with the dispatch center, involved emergency physicians, paramedics, or bystander who performed the emergency call. All patients have undergone targeted temperature management with target temperatures between 32 and 34 °C for 24 h. Rewarming was performed at a rate of 0.25–0.5 °C. Resuscitation related parameters were analyzed and reported according to the Utstein recommendations [23,24]. Patient demographics, co-medication, and concomitant diseases were obtained by chart review. Laboratory data were obtained from the ISO-credited central laboratory at the General Hospital of Vienna. Blood samples were obtained at admission, 12 h and 24 h after admission and at rewarming. Ethylendiaminetetraacetic acid (EDTA) anti-coagulated blood was centrifuged at 2000 g for 10 min. Plasma was obtained and stored at –80 °C until batch analysis. PCSK–9 concentrations were measured by enzyme-linked immunoassays in EDTA plasma (Circulex human PCSK–9 Elisa, MBL International, Woburn, MA, USA).

### 2.1. Definition of Multi-Organ Failure (MOF):

Multi-organ failure was defined as a failure of two or more organs within the first 48 h after admission. Acute kidney failure was defined as an elevation of serum- creatinine ≥ 0.5 mg/dL within the first 48 h. Acute shock liver was defined as an elevation of glutamate oxalacetate transaminase (GOT) and glutamate pyruvate transaminase (GPT) > 10–100 fold within the first 48 h.

Respiratory insufficiency was defined as > 0.35 FiO2 (= inspiratory oxygen fraction) and >0.1 mcg/kg/min epinephrine within the first 48 h.

### 2.2. Endpoints

The primary outcome was favorable neurologic function at day 30 after resuscitation, assessed using the cerebral performance category (CPC) 5-points scale (CPC 1–2 = favorable neurologic function vs. CPC 3–5 = a composite of unfavorable neurologic function and death). CPC was assessed by study fellows through structured face-to-face or telephone interviews with the patient, the relatives, treating physicians, or nursing home members.

30-day mortality was the secondary outcome.

We investigated the influence of PCSK–9 levels on these outcomes, but also associations with inflammatory and other resuscitation specific parameters.

### 2.3. Statistics

We present continuous data as median ± quartiles, and categorised data as absolute count and relative frequency. The Mann–Whitney *U* test was used to test for differences between two independent group medians. For categorised data we used a chi-squared test. Receiver operating characteristic curve (ROC) analysis was performed (i) to analyze specificity and sensitivity of PCSK–9 concentrations to predict neurological outcome and (ii) to identify the PCSK 9 level cut-off for optimal discrimination between favorable and unfavorable 30-day neurologic function (primary outcome). Additionally, we performed ROC-analysis for the secondary endpoint 30-day mortality. To estimate the association between PCSK–9 levels, potential confounders and primary outcome we used logistic regression analysis. We included covariables into the models based on clinical reasoning. Continuous variables were categorised and used as index variables. If applicable, we added a separate category for missing observations, otherwise no data-imputation for missing data was applied. We tested for linearity and interactions of the main effects using the likelihood ratio test. The log rank test was used to compare survival distributions between patients with PCSK–9 levels below and above the identified optimal cut-off at admission. Cox-regression analysis was used to estimate the association of PCSK–9 levels and the included co-variables with 30-day mortality and to assess potential confounding. The limited sample size precluded the calculation of multivariable regression models. Therefore, we calculated models only including sex and age to present adjusted odds (or hazard) ratios for these two factors. Moreover, we calculated multiple linear regression models including all covariables, because this procedure is more apt at dealing with small sample sizes (results are presented in the Appendix A). For data management and analyses we used Stata 14 (Stata Corp, College Station, Tx, USA) and IBM SPSS Statistics (Version 26, IBM Corporation, NY, USA). Generally, a two-sided *p*-value less 0.05 was considered statistically significant.

## 3. Results

Seventy-nine patients were included in this prospective study, of whom 43 (54%) had an unfavorable neurologic outcome and 26 patients (33%) died (Table 1). Baseline characteristics according to the PCSK-9 cutoff of the secondary endpoint 30-day mortality are presented in the Appendix A. 

At admission, PCSK-9 levels were significantly lower in patients with favorable than in those with unfavorable neurologic function (median 158 (quartiles: 124–225) ng/mL vs. 207 (174–259) ng/mL; *p* = 0.019). Overall PCSK-9 levels changed only marginally in the first 24 h, but increased significantly after rewarming (Figure 1). 

### 3.1. Receiver Operating Characteristic Curve Analysis

The optimal PCSK–9 level cut-off for the discrimination between favorable and unfavorable neurologic function was 165 ng/mL based on ROC analysis (area under the curve (AUC) = 0.67; 95% confidence interval (CI) 0.55–0.80, Figure 2). Sensitivity for this cut-off was 79% and specificity was 56%.

The optimal PCSK–9 level cut-off for the discrimination between 30-day survival and 30-day mortality was 180 ng/mL based on ROC analysis (AUC = 0.67, 95% CI 0.54–0.79, Figure 2) with a sensitivity of 81% and a specificity of 51%.

### 3.2. Neurologic Function

More patients with PCSK–9 concentrations < 165 ng/mL had a favorable neurologic function (69%) compared to patients having PSCK-9 concentrations ≥ 165 ng/mL (32%, *p* = 0.001).

In crude unadjusted analysis patients with PCSK–9 levels ≥ 165 ng/mL were more likely to have a favorable neurological outcome (odds ratio (OR) 4.72; 95% confidence intervals (95%CI) 1.76–12.66; *p* = 0.007, Appendix A). Likewise, elevated C-reactive protein (CRP) levels (≥0.5 mg/dL) at admission were more likely to have a CPC score of 3–5 (OR 3.29; 1.12–9.62; *p* = 0.03). Expectedly, patients with multi-organ failure (MOF) were also more likely to have an unfavorable neurological outcome (OR 21.0; 6.53–67.53, *p* < 0.001). PCSK–9 levels were not associated with the occurrence of MOF.

In multivariable analysis we included PCSK–9 levels, initial heart rhythm, no-flow interval, CRP levels, age and gender. The size of the sample did not allow for a multivariable analysis containing all co-variables simultaneously, but multivariable analyses including each relevant co-variable separately could indicate possible confounding by age (*p* = 0.041) or initial rhythm (*p* = 0.031) (Appendix A). We could not identify significant interactions. Statin intake may increase PCSK–9 concentrations [25], but in our study we did not observe any influence on PCSK–9 levels on the outcome.

Additionally, we performed a multivariable logistic regression analysis including only sex, age and PCSK–9 levels (≥165 ng/mL). The adjusted odds ratios were 4.46 (95%CI 1.61–12.38) for PCSK–9 and 1.04 (95%CI 1.00–1.08) for age, while sex was eliminated from the model.

### 3.3. Mortality

Thirty-day mortality was significantly higher in patients with PCSK–9 levels ≥180 ng/mL (18% vs. 46%; *p* = 0.006) (Figure 3).

In crude, unadjusted analysis, patients with PCSK–9 levels < 180 ng/mL had a hazard ratio (HR) of 3.52 (95%CI 1.33–9.36, *p* = 0.012). Expectedly, age (HR 1.04, 95CI% 1.01–1.06, *p* = 0.015), MOF (HR 6.38, 95%CI 2.21–18.43, *p* = 0.001), initial shockable rhythm (HR 0.29, 95%CI 0.14–0.62, *p* = 0.001), and CRP at admission (HR 1.22, 95%CI 1.07–1.39, *p* = 0.004) were also associated with mortality. Since the limited sample size precluded a multivariable analysis including all relevant covariables simultaneously, similarly to the above-mentioned analysis, we included the relevant co-variables separately, to test for potential confounding. We cannot exclude confounding by age (*p* = 0.025), initial shockable rhythm (*p* = 0.007) and elevated CRP levels (*p* = 0.02). Additionally, we calculated a limited multivariable Cox regression model including only sex, age and PCSK–9 levels (</≥180 ng/mL). However, only age (HR 1.04; 95%CI 1.01–1.08, *p* = 0.009) and PCSK–9 levels (HR 3.10, 95% CI 1.16–8.28) remained in the model, while sex was eliminated in the backward elimination procedure.

### 3.4. Additional Analyses

Since both, CRP levels and PCSK–9 levels were associated with neurologic function, we categorized patients into four groups: low CRP and low PCSK–9, high CRP and low PCSK–9, low CRP and high PCSK–9, high CRP and high PCSK–9. For PCSK–9 the respective cutoffs for neurologic function (165 ng/mL) and for 30–day mortality (180 ng/mL) were used. For CRP levels no clear cutoff was identifiable in ROC analysis and, therefore, the upper limit of normal (reference range < 0.5 mg/dL) was used. There was a significant difference in the distribution of favorable and unfavorable neurologic outcome (*p* = 0.007) and 30–day mortality (*p* = 0.005) over these four groups (Table 2).

In a post-hoc analysis we analyzed whether neuron-specific enolase or the neuron-specific protein S100, which were available for 52 and 51 patients (measured between 24 and 48 h after cardiac arrest), respectively, correlate with PCSK–9 levels. However, no such correlation could be identified.

## 4. Discussion

In this study we analyzed PCSK–9 kinetics in patients successfully resuscitated from out of hospital cardiac arrest and hypothesized an association with neurologic outcome and 30-day mortality. We found that high PCSK–9 levels (≥165 ng/mL) at admission were associated with unfavorable 30-day neurologic function (OR 4.72; 95%CI 1.76–12.66; *p* = 0.007) and an increased 30-day mortality (PCSK–9 ≥180 ng/mL, 15% vs. 46%, hazard ratio 3.52 (95%CI 1.33–9.36, *p* = 0.012). Moreover, CRP concentrations were lower in patients with PCSK–9 concentrations < 165 ng/mL at admission and after 12 h.

Loss-of-function mutations but also low PCSK–9 concentrations were associated with lower mortality in septic patients, while high PCSK–9 levels were associated with organ dysfunction [11,14,26]. The assumed mechanism is a more rapid detoxification of bacterial lipids via higher LDL-R expression and consequently a reduced inflammatory response [11,14]. Given the similarities between sepsis and PCAS [27,28,29], we hypothesized that low PCSK–9 levels may be associated with favorable outcome after out-of-hospital cardiac arrest and indeed our findings confirm this hypothesis. Moreover, recently Lee et al. reported that in patients after cardiopulmonary resuscitation, total cholesterol, low-density lipoprotein, as well as high-density lipoprotein differed between those with favorable or unfavorable outcome, whereas especially high-density lipoprotein and total cholesterol were associated with clinical outcomes after regression analysis [30].

Based on the underlying mechanism, we expected lower levels of inflammatory biomarkers in patients with low PCSK–9 levels. However, we only observed lower CRP levels in the first 12 h after admission. Of note, the causes for inflammation after CPR are multifactorial including increased intestinal permeability, but also aspiration pneumonia, ischemic events, comorbidities, etc. Also, therapeutic interventions including temperature management or pharmacological treatment may affect the inflammatory response, which in sum may explain the missing association at later time points. Similarly, to our findings Vaahersalo et al. reported that interleukin-6 levels only at admission, but not at later time points predicted neurologic outcome [27].

Furthermore, recent data suggest that PCSK–9 could be involved in the release of pro-inflammatory cytokines after hypoxia and reperfusion [22]. Moreover, PCSK–9 also interacts with LOX-1, the receptor for oxidized LDL, which is upregulated in ischemic hearts [21,22]. Low levels of PCSK–9 (or inhibition of PCSK–9) could therefore exert beneficial effects–Of note, all patients included in this analysis had an acute coronary syndrome as cause of cardiac arrest. It is therefore possible, that in this selected cohort the associations of PCSK–9 levels with outcomes are especially pronounced. Thus, our findings should be confirmed in other cardiac arrest populations, including patients with non-cardiac cause of cardiac arrest.

PCSK–9 levels changed only marginally during the first 24 h, but increased significantly until rewarming in both groups, while CRP levels increased constantly. The increase in PCSK–9 levels is most likely caused by the inflammatory response [2]. Of note, during experimental endotoxemia in healthy volunteers PCSK–9 levels initially decreased by approximately 40% before they returned to baseline [15]. This observation ended after 24 h, but there were no increases in PCSK–9 in that period. Any possible increases in that model would take at least 24 h, which corresponds well with our results.

Astonishingly, the marked difference in neurologic outcome in patients having PCSK–9 concentrations < 165 ng/mL (or < 180 ng/mL for 30–day mortality) cannot be explained by differences in the “classical” CPR variables: bystander status, basic life-support, no-flow, low-flow intervals, time to sustained ROSC, and initial heart rhythm differed only marginally. It is noteworthy that only patients with sustained ROSC at admission were included in the trial.

In multivariate analysis none of the chosen factors remained significant. However, this may also be caused by the small sample size in the study and should be re-evaluated in a larger sample.

Based on our results, low PCSK–9 levels may have beneficial effects on the neurologic outcome after CPR. Thus, early pharmacologic inhibition of PCSK–9 may be an interesting treatment option to improve neurologic survival. Further research may be warranted to elucidate its therapeutic potential.

Limitations: this is a cohort study with its inherent limitations. The population was selected based on the inclusion and exclusion criteria, and we cannot exclude a potentially associated selection bias. The effects of targeted temperature management on PCSK–9 levels and metabolism are unknown. The sample size is limited and analysis in larger populations to confirm our results is warranted. We did not measure endotoxins or other lipoprotein levels. All patients were treated with target temperature management, which may influence PCSK–9 levels.

## 5. Conclusions

In conclusion, lower PCSK9 levels at admission were associated with favorable neurologic outcome after CPR.

## Figures and Tables

**Figure 1 jcm-09-02606-f001:**
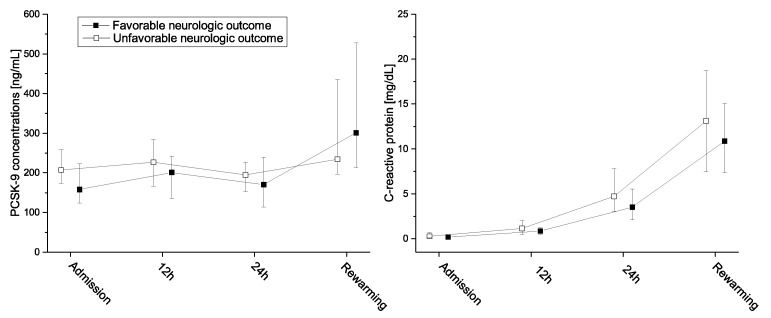
PCSK–9 (left panel) and CRP-levels (right panel) for favorable and unfavorable 30-day neurologic function. Data are median ± IQR. PCSK–9, proprotein convertase subtilisin/kexin type 9.

**Figure 2 jcm-09-02606-f002:**
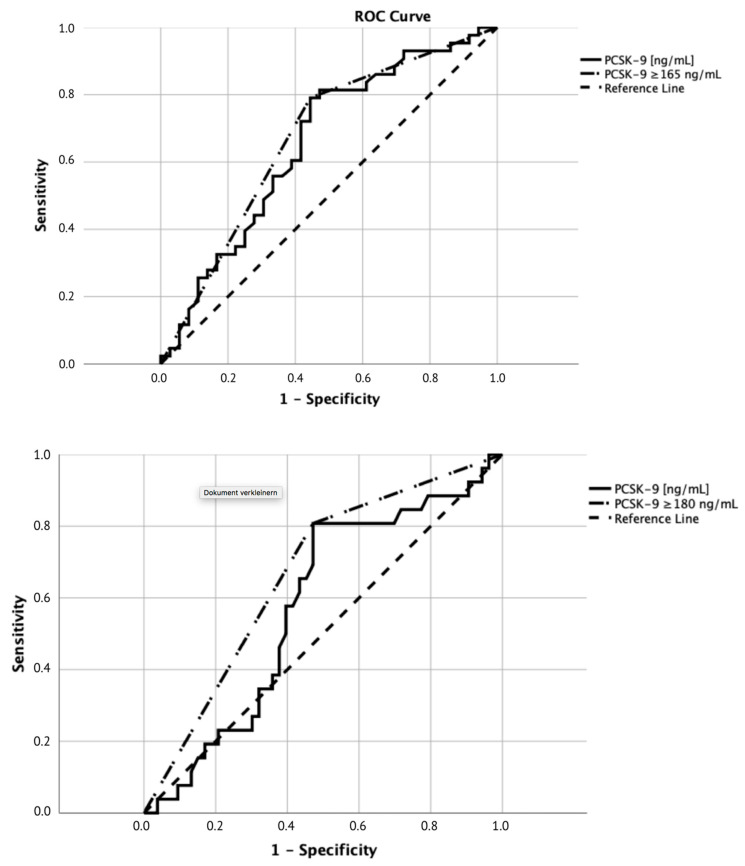
Receiver operating characteristic curve (ROC) analyses. Upper panel: ROC analysis for unfavorable neurologic outcome, the optimal cutoff for PCSK–9 was <165/≥165 ng/mL. The area under the curve (AUC) was 0.67 (95% confidence intervals 0.55–0.80). Sensitivity for this cutoff was 79%, while specificity was 56%. Lower panel: ROC analysis for 30-day mortality, the optimal cutoff for PCSK–9 was <180/≥180 ng/mL. The AUC was 0.67 (95% confidence intervals 0.54–0.79).

**Figure 3 jcm-09-02606-f003:**
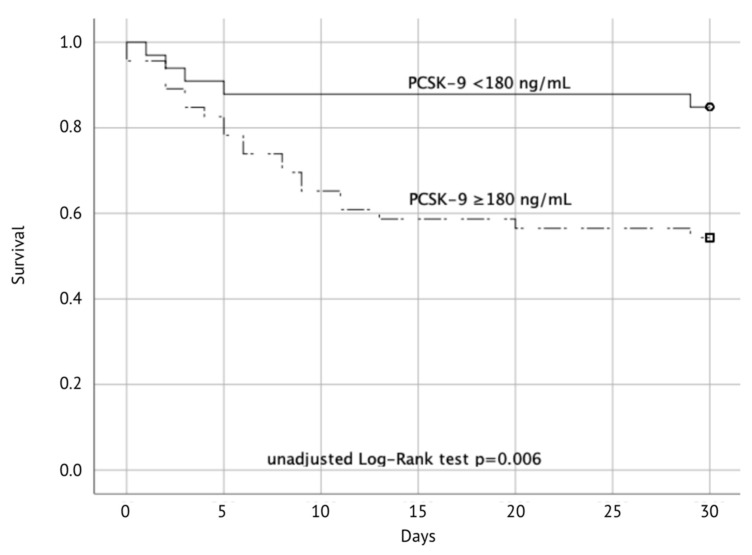
Secondary outcome: Cumulative probability of survival to day 30 after successful resuscitation according to PCSK–9 levels at admission (PCSK–9 < 180 ng/mL vs. ≥ 180 ng/mL).

**Table 1 jcm-09-02606-t001:** Baseline Characteristics: Categorical data are presented as counts and percentages, continuous data as medians and interquartile ranges (IQRs).

Baseline Characteristics	Total (*n* = 79)	PCSK-9–Levels (ng/mL)		
		<165 (*n* = 29)	≥165 (*n* = 50)	*p*
Gender, male *n* (%)	61 (77)	25 (86)	36 (72)	0.149
Age, years (IQR)	59 (46–69)	56 (44–67)	59 (51–70)	0.222
Concomitant diseases, *n* (%)				
Hyperlipidemia	20 (25)	5 (17)	15 (30)	0.212
DM II	16 (20)	7 (24)	9 (18)	0.516
Coronary artery disease	16 (20)	7 (24)	9 (18)	0.516
Hypertension	31 (39)	13 (45)	18 (36)	0.442
Smoker	26 (33)	9 (31)	17 (34)	0.788
COPD	8 (10)	3 (10)	5 (10)	0.961
PAD	6 (8)	0 (0)	6 (12)	0.054
Statine-use, *n* (%)	14 (18)	3 (10)	11 (22)	0.194
CPC 1/2, *n* (%)	36 (46)	20 (69)	16 (32)	0.002
Witnessed, *n* (%)	64 (81)	22 (76)	42 (84)	0.276
BLS, *n* (%)	55 (70)	19 (66)	36 (74)	0.375
Initial shockable rhythm, *n* (%)	59 (75)	22 (76)	37 (74)	0.855
Time to sustained ROSC, min (IQR)	25 (17–43)	20 (13–41)	29 (20–43)	0.056
No-flow time, min (IQR)	0 (0–3)	0 (0–3)	0 (0–3)	0.846
Low-flow time, min (IQR)	25.0 (16.0–40.5)	20 (10–38)	27 (20–41)	0.064
30-day survival, *n* (%)	53 (67)	24 (83)	29 (58)	0.025
Core body temperature (admission), °C (IQR)	35.3 (34.8–35.8)	35.5 (34.9–35.9)	35.2 (34.7–35.7)	0.314
Blood gas values (admission)				
pH (IQR)	7.2 (7.0–7.1)	7.2 (6.9–7.3)	7.2 (7.1–7.2)	0.782
Lactate, mmol/L (IQR)	7.6 (5.2–10.0)	7.2 (4.4–10.6)	7.6 (6.1–9.9)	0.625
Laboratory values (admission)				
Hemoglobin, g/dL (IQR)	13.9 (12.4–15.3)	14.1 (12.8–15.4)	13.8 (12.2–15.2)	0.444
Total Platelet count, G/L (IQR)	205 (169–245)	225 (179–250)	202 (163–245)	0.502
Leukocytes, G/L (IQR)	14.3 (10.5–18.9)	14.9 (8.7–19.1)	14.3 (11.0–18.4)	0.499
Troponin-T, ng/L (IQR)	64 (32–244)	130 (38–467)	54 (27–192)	0.153
Prothrombin-time, % (IQR)	77 (65–91)	72 (59–83)	81 (66–91)	0.184
TG, mg/dL (IQR)	125 (81–173)	115 (76–154)	133 (83–179)	0.292
Creatinin, mg/dL (IQR)	1.1 (0.9–1.3)	1.2 (0.9–1.3)	(0.9–1.3)	0.577
Albumin, g/L (IQR)	38 (34–40)	39 (36–41)	37 (33–41)	0.048
Cholinesterase, kU/L (IQR)	6.7 (5.4–8.0)	6.8 (5.1–8.1)	6.7 (5.5–7.8)	0.718
Total cholesterol, mg/dL (IQR)	163 (133–201)	160 (139–182)	166 (127–212)	0.897
ASAT (GOT), U/L (IQR)	135 (78–226)	135 (75–201)	137 (83–264)	0.600
ALAT (GPT), U/L (IQR)	112 (56–200)	107 (66–164)	127 (54–210)	0.640
Laboratory values				
PCSK–9 (admission), ng/mL (IQR)	193 (145–239)	126 (115–147)	223 (194–266)	0.000
PCSK–9 (12 h), ng/mL (IQR)	210 (151–267)	156 (127–209)	239 (204–298)	0.000
PCSK–9 (24 h), ng/mL (IQR)	187 (198–457)	145 (108–213)	202 (178–244)	0.002
NSE (24 h), µg/L (IQR)	33.1 (21.4–54.4)	29.6 (20.3–44.8)	34.7 (22.1–57.6)	0.370
S-100 (24 h), µg/L (IQR)	0.15 (0.09–0.36)	0.10 (0.10–0.16)	0.17 (0.11–0.52)	0.007
CRP (admission), mg/dL (IQR)	0.2 (0.1–0.6)	0.2 (0.1–0.3)	0.3 (0.1–0.8)	0.045
CRP (12 h), mg/dL (IQR)	0.99 (0.6–1.6)	0.8 (0.5–1.1)	1.3 (0.7–2.1)	0.029
CRP (24 h), mg/dL (IQR)	4.2 (2.8–6.1)	3.8 (2.4–5.4)	4.4 (2.8–7.7)	0.270

CPC, cerebral performance category; DM II, diabetes mellitus II; COPD, chronic obstructive pulmonary disease; PAD, peripheral arterial disease; BLS, basic life support; ROSC, return of spontaneous circulation; PCSK–9, proprotein convertase subtilisin/kexin type 9; TG, triglycerides; ASAT, aspartate aminotransferase; GOT, glutamate oxalacetate transaminase; ALAT, alanine aminotransferase; GPT, glutamate pyruvate transaminase; CRP, C-reactive protein; NSE, neuro-specific enolase.

**Table 2 jcm-09-02606-t002:** Outcome analysis for groups according to PCSK–9 and C-reactive protein levels.

	Low CRP Low PCSK–9	High CRP Low PCSK–9	Low CRP High PCSK–9	High CRP High PCSK–9
Unfavorable neurologic function (CPC 3–5)	28% *n* = 25	50% *n* = 4	61% *n* = 31	79% *n* = 19
30-day mortality	11% *n* = 28	40% *n* = 5	36% *n* = 28	61% *n* = 18

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
