# Peer review of "Low PCSK-9 levels Are Associated with Favorable Neurologic Function after Resuscitation from out of Hospital Cardiac Arrest"

_jcm, 2020, doi:10.3390/jcm9082606_

Round 1

Reviewer 1 Report

The authors evaluate the relationship between neurologic function, survival and PCSK9 in resuscitated patients. This issue raises the great interest and provides important information to readers. However, there are some limitations to be published.

  1. Authors There is no reasonable explanation or process to draw PCSK9 level cut-off for the discrimination between favorable and unfavorable. The optimal cut off level should be supported based on statistical or clinical evidence.

  1. Supplementary figure 1 is difficult to understand. Table format should be more organized than submitted one.

  1. It seems that baseline characteristics based on CPC is not necessary. PCSK9 is the main interest, not CPC.

  1. Authors suggest that CRP levels tended to increase with PCSK9 concentrations. Correlation analysis between PCSK9 and CRP at each follow up is better to demonstrate results.

  1. Several causes raise the cardiac arrest and the causes of arrest might affect PCSK9 level. It would be better to be included in baseline characteristics, while causes are frequently unknown.

  1. While I agree limited sample size cause difficulty to do multivariate logistic or Cox regression analysis, it is very important to persuade the readers. I recommend multiple linear regression analysis using cerebral performance score as dependent variable. Linear regression analysis is useful to fulfil the sample size. In the same way, multivariate Cox-regression is positively necessary. However, this analysis is difficult to do, given the small event of survival. Therefore, adjusted hazard ratio with minimal covariates such as age and sex is helpful to demonstrate results.

  1. The CRP is most prevalent marker to evaluate the resuscitated patients in the clinical setting. To strengthen the clinical usefulness of PCSK9, it seems better that patients are categorized into four groups based on PCSK9 and CRP level in association with neurologic score.

Reviewer 2 Report

This topic is interesting area for understanding of post-cardiac arrest syndrome. 

I have a few questions. 

  1. this study has a small number of participants. It is lack of the numbers of revealing the relations of PCSK-9 and neurologic outcome.
  2. If you have  any time-serial biomarkers such as NSE, S-100 or any levels of lipid profile, corresponding to the value of PCSK-9 at each time?
  3. You have to show the results of ROC curve, sensitivity and specificity for setting the level cut-off of PCSK-9 for the discrimination between favorable and unfavorable neurologic function as 180 ng/ml.
  4. You have to show the results of odds ratio, that was described in the abstract? (ABSTRACT : In patients with PCSK 9 levels ≤180 ng/ml,
    25 the odds of favorable outcome were 3.62 fold (95% CI 1.41-9.26; p=0.007) higher compared to those 26 with PCSK 9 levels >180 ng/ml.)
  5. The level of PCSK-9 at 12 hour, and 24 hour, may be influenced by TTM and any inflammatory response like pneumonia, or any infectious conditions. It would be better to show the patients' diseases, that was related to inflammatory conditions and initial laboratories like CBC, creatinine, and liver function test in Appendix.
  6. You concluded 'lower PCSK9 levels at admission were associated with favorable neurologic outcome after CPR.' I think that the level of PCSK 9 is important to the trends of level of PCSK 9  as well as the initial level. 

Round 2

Reviewer 1 Report

I appreciate your effort and exertion to improve manuscript quality.